# Predicting lymph node metastasis from primary tumor histology and clinicopathologic factors in colorectal cancer using deep learning

Justin D. Krogue [1✉], Shekoofeh Azizi [2], Fraser Tan[1], Isabelle Flament-Auvigne[3], Trissia Brown[3], Markus Plass [4], Robert Reihs [4], Heimo Müller [4], Kurt Zatloukal [4], Pema Richeson[5], Greg S. Corrado[1], Lily H. Peng[1], Craig H. Mermel[1], Yun Liu [1], Po-Hsuan Cameron Chen [1], Saurabh Gombar[5], Thomas Montine[5], Jeanne Shen[5], David F. Steiner [1,6] & Ellery Wulczyn[1,6]

## Abstract

**Background** Presence of lymph node metastasis (LNM) influences prognosis and clinical decision-making in colorectal cancer. However, detection of LNM is variable and depends on a number of external factors. Deep learning has shown success in computational pathology, but has struggled to boost performance when combined with known predictors.

**Methods** Machine-learned features are created by clustering deep learning embeddings of small patches of tumor in colorectal cancer via k-means, and then selecting the top clusters that add predictive value to a logistic regression model when combined with known baseline clinicopathological variables. We then analyze performance of logistic regression models trained with and without these machine-learned features in combination with the baseline variables.

**Results** The machine-learned extracted features provide independent signal for the presence of LNM (AUROC: 0.638, 95% CI: [0.590, 0.683]). Furthermore, the machine-learned features add predictive value to the set of 6 clinicopathologic variables in an external validation set (likelihood ratio test, $p < 0.00032$; AUROC: 0.740, 95% CI: [0.701, 0.780]). A model incorporating these features can also further risk-stratify patients with and without identified metastasis ($p < 0.001$ for both stage II and stage III).

**Conclusion** This work demonstrates an effective approach to combine deep learning with established clinicopathologic factors in order to identify independently informative features associated with LNM. Further work building on these specific results may have important impact in prognostication and therapeutic decision making for LNM. Additionally, this general computational approach may prove useful in other contexts.

## Plain language summary

When colorectal cancers spread to the lymph nodes, it can indicate a poorer prognosis. However, detecting lymph node metastasis (spread) can be difficult and depends on a number of factors such as how samples are taken and processed. Here, we show that machine learning, which involves computer software learning from patterns in data, can predict lymph node metastasis in patients with colorectal cancer from the microscopic appearance of their primary tumor and the clinical characteristics of the patients. We also show that the same approach can predict patient survival. With further work, our approach may help clinicians to inform patients about their prognosis and decide on appropriate treatments.

[1] Google Health, Palo Alto, California, USA. [2] Google Research, Brain Team, Toronto, ON, Canada. [3] Google Health via Vituity, Emeryville, CA, USA. [4] Medical University of Graz, Graz, Austria. [5] Department of Pathology, Stanford University School of Medicine, Stanford, California, USA. [6] These authors jointly supervised this work: David F. Steiner, Ellery Wulczyn. ✉email: justin.d.krogue@gmail.com

Colorectal cancer (CRC) is the third most common type of cancer in the United States and the third-leading cause of cancer-related deaths[1]. The presence of lymph node metastasis (LNM) is a key factor in the prognosis and management of CRC. For cases without LNM (stage I/II disease), treatment is typically via surgical resection alone, whereas the presence of LNM (stage III disease) is an indication for adjuvant chemotherapy[2,3].

Unfortunately, the current diagnosis of LNM is imperfect. To identify metastasis within a lymph node, the affected lymph node must first be contained in the surgical resection, identified in the surgical specimen for pathologic evaluation, and then the metastasis within must be recognized on histologic sectioning and review; all of these processes are subject to substantial variability[4]. For example, surgeons vary in their lymph node yield depending on the margin size taken and whether additional lymph node dissection is done[3], and the number of nodes identified within a surgical specimen is associated with the amount of time taken by the pathologist[5,6]. After finding the lymph nodes, identifying LNM can further depend on specimen processing and careful evaluation. Cutting multiple sections from the node increases detection of LNM[7] and small metastases (such as isolated tumor cells or micrometastases) may be missed[8]. Immunohistochemistry stains and molecular techniques may enhance the yield but are variably performed due to cost, availability, and workflow challenges[3]. Consistent with these diagnostic challenges, patients with node-negative disease still have 5-year mortality of 20–30%, which is thought to be due largely to undiagnosed lymph node involvement[3,5]. Therefore, better identification of LNM would have significant ramifications for patient prognosis and therapeutic management.

While prior work has demonstrated associations of established histologic factors with LNM, including submucosal involvement, tumor budding, lymphovascular invasion, tumor growth pattern, fibrotic stroma, and various IHC markers[9–11], no system currently exists in practice to utilize these features to make an accurate risk estimate for LNM. Deep learning via convolutional neural networks has shown great success in medical image analysis, including prediction of clinical outcomes across many cancer types from pathological images via weakly-supervised deep learning[12–16]. For LNM prediction, deep learning approaches have recently shown promise as well, but to this point have struggled to show improved predictive power over models using baseline clinicopathologic variables in external validation[17–20]. Without controlling for established histologic associations, a deep learning system may directly learn to predict the known features, limiting overall performance relative to an approach that incorporates both prior-known and deep learning-extracted previously unknown features.

In this study we propose a method of controlling for known variables while selecting machine-learned features, with the aim of developing a combined predictive model that maximizes generalizable performance while being inherently interpretable. We hypothesize that such a system will achieve better performance than a model trained on known clinicopathologic variables or deep learning based features alone. To our knowledge, this method of feature generation/selection has not been done before, and we demonstrate that this does indeed provide a performance boost over known baseline variables that generalizes to an external dataset. Finally, we explore the ability of different deep learning pre-training regimens to generate relevant machine-learned features, including supervised pretraining using natural images vs self-supervised pre-training using histologic images.

## Methods

**Data cohorts**. This retrospective study utilized de-identified, digitized histopathology slides of primary colorectal samples and clinicopathologic metadata from colorectal cancer cases from the BioBank at the Medical University of Graz (MUG)[21] and from Stanford University (SU). Slides were scanned using a Leica Aperio AT2 scanner at 20X magnification (0.5 μm/pixel). Institutional Review Board approval for this study was obtained from MUG (Protocol no. 30-184 ex 17/18) and SU (Protocol no. 46762). The need for informed consent was waived as the research was deemed to involve no more than minimal risk to the subjects, and could not be practically carried out with informed consent given the historical nature of the retrospective datasets. Clinicopathologic metadata including pathologic TNM staging, age, sex, and tumor grade were extracted from de-identified clinical and pathology reports. When indicated in the report, presence of lymphatic invasion and venous invasion were also extracted. Only cases with complete clinicopathologic metadata for TNM staging, age, sex, tumor grade were included in this study. Patient characteristics of these cohorts are reported in Supplementary Data 1.

Cases from MUG comprise archived stage II and stage III colorectal cases from 1984 to 2013. Cases from 1984-2007 were used for model development and feature selection (divided into training and tune sets) and cases from 2008-2013 were used as a temporal validation set[22–24]. In the event of multiple cases for a given patient, only the primary resection was included. Cases from SU comprise all available archived stage II and III colorectal cancer cases and a random sample of available stage I and IV cases from 2007 to 2018 (one case per patient). The SU cases were used for external validation.

All slides underwent manual quality assurance review by pathologists to confirm the stain, tissue, and specimen type. Only hematoxylin and eosin (H&E) stained slides with colorectal tissue from resection specimens were used in this study. Slides containing sections from lymph nodes or large lymphoid aggregates were excluded.

Notably, as per colorectal cancer staging definitions[4], stage II cases are node negative and represent only T3 or T4 cases, while stage III cases are node positive and can include cases of any T category. Hence, in a cohort of stage II and stage III cases, T1 and T2 cases will all be node positive. To avoid learning a spurious association between node positivity and T1 or T2, we excluded T1 and T2 cases from the MUG cohorts. For evaluation using the external datasource (SU), we established a similar cohort (stage II/III, T-category T3/T4) which we refer to as external validation set 1a. The full SU cohort containing stage I-IV cases and all T-categories is referred to as external validation set 1b.

The final dataset used for model development also involved the inclusion criteria of requiring that the WSIs were scanned after a specific date. This stems from the fact that training/tune set cases for MUG were scanned with the majority of stage II cases preceding stage III in scan date. Unfortunately, in early model development, we discovered that our model learned this spurious association between scan date and stage, as even for LNM negative cases only (stage II), the model predicted higher probability of node positivity for cases after a certain scan date. Despite reviewing our scanning operation notes, investigating scanning metadata, and examining scanned slides, we could not elucidate what had changed. Additionally, there was no difference in clinicopathologic variables (e.g., T-category, grade, etc) between cases before and after this date that may explain the difference in average model output. Therefore, for model development, we dropped all cases that were scanned before this date, and subsampled stage III cases to account for the subsequent class imbalance resulting from the relative decrease in stage II cases. This did not affect cases in the internal or external validation sets.

The MUG data cohorts were detailed previously[16]; a STARD diagram detailing inclusion and exclusion criteria for the SU cohorts is available as Supplementary Fig. S8.

**Region of Interest Selection**. Region of interest (ROI) masks to utilize predominantly tumor regions for model development and inference were generated as described in a previous work[16]. Briefly, a convolutional neural network (CNN) was developed in a patch-based supervised learning approach using whole slide images (WSIs) from pathologist-annotated colorectal slides. This CNN takes as input a small patch from a WSI and predicts the presence of tumor in this patch. We then compare the prediction of the model for this patch with the ground truth obtained via the pathologist-labeled tumor segmentation mask for the slide. This model achieved an AUC of 0.985 with sensitivity of 0.940 and specificity of 0.950 on a held out test-set of 44 slides (6,866,573 patches) from the Medical University of Graz and an AUC of 0.949 with sensitivity of 0.880 and specificity of 0.872 on a held out test set of 86 slides (10,005,507 patches) from the public TCGA COAD data set. For more details, see ref. [15]. Tumor masks were generated by running the tumor detection model over all tissue containing patches in a slide in a sliding-window fashion (Fig. 1a). ROI masks were generated by denoising and dilating the tumor masks to capture both tumor regions and approximately 0.5 mm of bordering non-tumor regions (approximately 2 patch-widths) immediately adjacent to the tumor.

**Feature Generation**. Candidate case-level machine-learned features for predicting LNM were generated by computing the percentage of patches in each case belonging to each of K different histologic clusters. Clusters were generated by applying the k-means algorithm to embeddings of a sample of 113,984 patches, where an embedding represents the feature vector obtained after passing a patch through a convolutional neural network (CNN) (Fig. 1a). This set of patches was constructed by randomly sampling 64 patches at 10X magnification from each case in the training set. Patch size was determined based on the design model architecture (289 × 289 pixels for Graph-Rise, 224 × 224 pixels for ResNet-based models at 1 micron per pixel). For the primary analysis, embeddings were generated via Graph-Rise[25], a CNN trained on natural images to predict image similarity, an approach consistent with prior work on pathology image search[25,26] and survival prediction[16]. Alternative embedding models including a BiT[27] model pre-trained on ImageNet and a self-supervised SimCLR[28] model pre-trained on The Cancer Genome Atlas (TCGA) were explored in secondary analyses (see Supplementary Methods).

**Feature selection**. Given a set of K candidate machine-learned features, a subset of machine-learned features was selected for inclusion in an LNM prediction model that combines both clinicopathologic variables and machine-learned features (Fig. 1b). Forward stepwise selection was employed for machine-learned feature selection with the baseline clinicopathologic variables included in the model throughout the process. In other words, we started with a multivariable logistic regression model for LNM that included only the set of known baseline variables. Candidate machine-learned features were then iteratively selected for inclusion in the multivariable logistic regression model. In each iteration, the candidate machine-learned feature that gave the largest increase in performance (AUROC) when added to the model was selected. We measured performance on the development set over different values of K clusters (10, 25, 50, 100, 200), and for an increasing number of selected machine-learned features (1–10). The optimal configuration of K = 200 and 5 selected machine-learned features was chosen based on development set performance and the observation of diminishing returns after selecting more than 5 machine-learned features.

The goal of this selection process was to identify a subset of machine-learned features that are associated with LNM after controlling for known clinicopathologic variables by including them in the multivariable model. The statistical evaluations used to evaluate this approach are described next.

**Statistical evaluation of features**. The association of the 5 machine-learned features with LNM after controlling for known clinicopathologic variables was evaluated in three different ways. Our first approach and planned primary analysis consisted of a likelihood ratio test comparing a null logistic regression model containing just the baseline clinicopathologic variables (age, sex, tumor grade, T-category, lymphatic invasion and venous invasion) to an alternative logistic regression model containing the baseline clinicopathologic variables and the 5 selected machine-learned features. In the second approach, we fit multivariable logistic regression models including both machine-learned features and known clinicopathologic variables and evaluated the odds ratios associated with the machine-learned features. In these first two approaches (which are both analytical in nature instead of predictive), the logistic regression models were fit on the validation set being evaluated. Lastly, we evaluated the gain in AUROC on the validation sets by adding the 5 selected machine-learned features to a baseline logistic regression model of clinicopathologic variables trained on the development set (e.g., a predictive logistic regression model) (Fig. 1c).

To further interpret histologic patterns captured by the machine-learned features, we sampled patches from different cases close to the cluster centroid for each of the top 5 machine-learned features (Fig. 2 and Supplementary Fig. S2–S6). Two board-certified pathologists reviewed these patches to provide a brief characterization of each machine-learned feature (Supplementary Table S2).

**Reporting summary**. Further information on research design is available in the Nature Portfolio Reporting Summary linked to this article.

## Results
Digitized histopathology slides of primary tissue as well as clinicopathologic metadata from two institutions' colorectal cancer patients were used in this study (Supplementary Data 1). From the first institution, cases from 1982 to 2007 were used for model development, while cases from 2008 to 2013 were used for temporal validation. Cases from the second institution were only used for external validation. For this external validation institution, we will refer to the cohort that contains only stage II and III cases as external validation set 1a and the full cohort containing stage I-IV cases as external validation set 1b.

We utilized a deep learning model to develop candidate machine-learned histologic features of colorectal cancer and selected 5 features that complement existing clinicopathologic variables for predicting LNM. In the pre-planned, primary analysis, we observed a significant improvement in the goodness of fit when adding the 5 machine learned features to a multivariable logistic regression model for LNM prediction containing 6 clinicopathologic variables in external validation set 1a. ($p = 0.00032$; likelihood ratio test; Table 1). Similar results were observed in the temporal validation set and external validation set 1b ($p < 0.0001$ for both, Table 1).

In multivariable regression analysis (Table 2) in which both clinicopathologic variables and machine learned features were

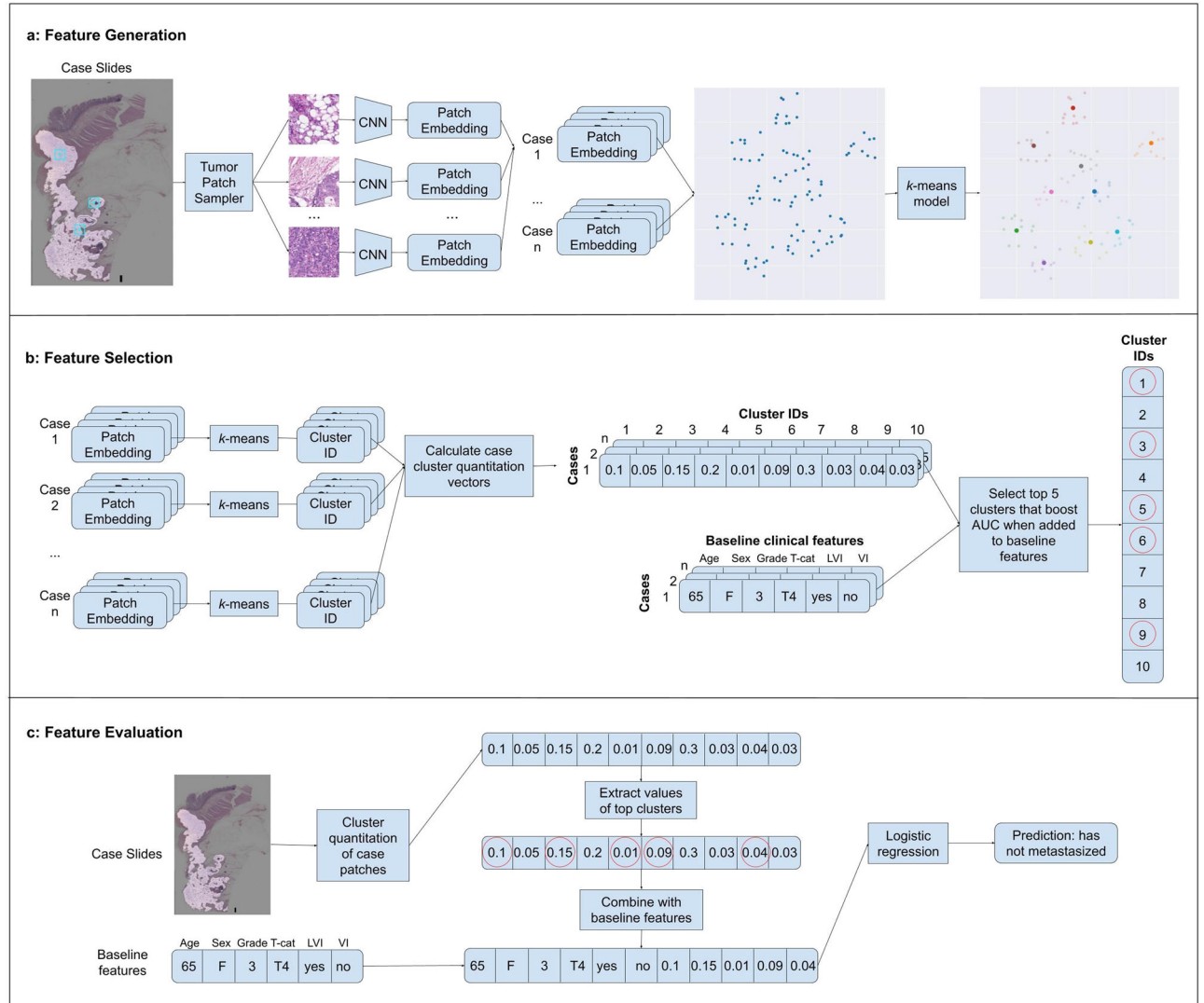

**Fig. 1 Overview of model development. a** Feature Generation: patches are sampled at random from regions containing tumor of a given case in the training set, and each patch is passed through a CNN to obtain an embedding vector. A k-means algorithm is then fit on these embeddings. Note: for demonstration purposes in this example only 10 cluster centroids were placed; in our actual model we fit 200 clusters on the patch embeddings. **b** Feature Selection: all patch embeddings from a case are run through the trained k-means model and are assigned a cluster id, and the fraction of patches in a case assigned to each cluster is computed (case-level cluster quantitation vector). This is repeated for all cases in the training set. The top 5 clusters are chosen to maximize AUROC in a greedy stepwise forward selection on the training set when combined with baseline clinical features in a logistic regression model. **c** Feature Evaluation: a cluster quantitation vector is computed for a case to be evaluated, and the cluster quantitations for the pre-selected top cluster ids are concatenated to the baseline clinical features. This case-level combined feature vector is then fed through a logistic regression model to obtain a prediction.

included, the odds ratios for two of the five individual machine-learned features were significant in each validation set, but the two features differed by validation set (features #1 and #3 in the temporal validation set; features #3 and #4 in the external validation sets).

We also evaluated the added predictive value of the machine-learned features by computing the increase in area under the receiver operating characteristic (AUROC) when adding the machine-learned features to a logistic regression model containing the clinicopathologic variables. These models were trained on the development set and evaluated on the validation sets. For external validation set 1a, addition of the machine learned features increased the AUROC by 0.024 [95% CI: −0.001, 0.048], from 0.716 [95% CI: 0.673, 0.759] using the baseline variables alone to 0.740 [95% CI: 0.697, 0.781] when also including the machine-learned features (Table 3). Similar improvements were seen for the temporal validation set and external validation set 1b

(Table 3). Receiver operating characteristic curves with AUROCs for these approaches are shown in Supplementary Figure S1. To evaluate the effect of model architecture and type of pretraining, we repeated this analysis for ResNet50 models that had either undergone supervised training via BiT on ImageNet[27] or self-supervised training via SimCLR[28] on The Cancer Genome Atlas, and found that all approaches gave similar results (Supplementary Table S3). We also report sensitivity, specificity, positive predictive value (PPV), and negative predictive value (NPV) for each method using an operating threshold by selecting the value that maximized the harmonic mean of sensitivity and specificity (Supplementary Table S4).

Given the potential clinical dilemma regarding use of adjuvant chemotherapy in stage II CRC patients, we also evaluated the ability of the same five machine-learned features to provide risk stratification for disease-specific survival (DSS). This analysis involved a logistic regression model trained on the machine-

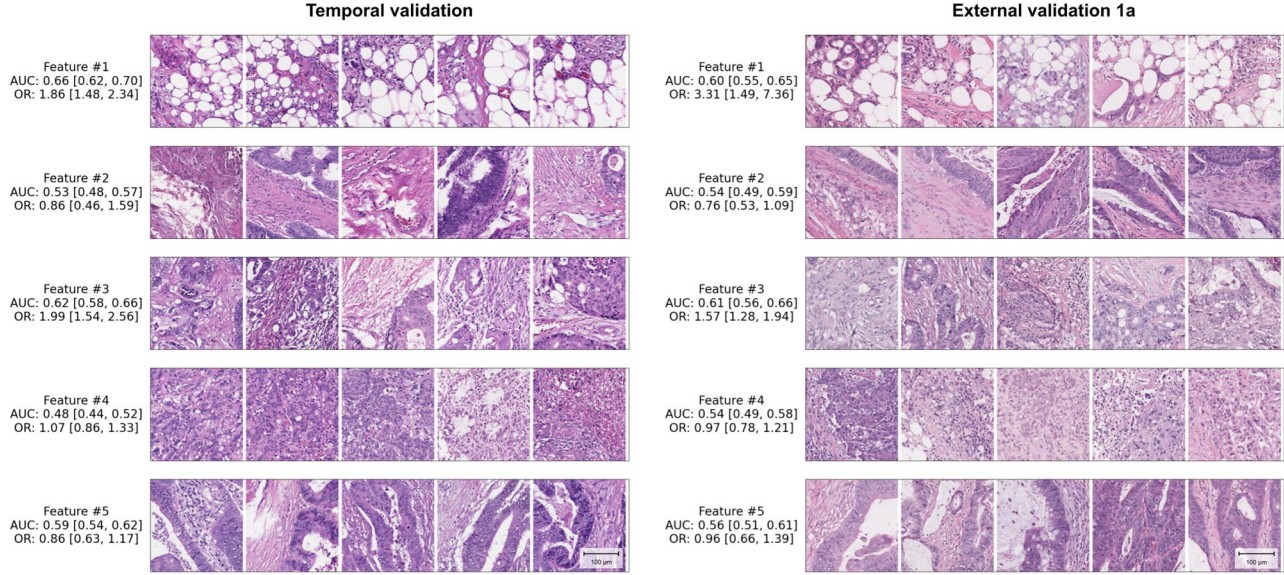

**Fig. 2 Representative patches of machine-learned features associated with lymph node metastasis.** Univariable AUROCs, univariable odds ratios (OR) and sample patches for the top 5 machine-learned features are shown. Patches representing the same machine-learned features are shown for the temporal validation set and the external validation set 1a, respectively. Patches selected here are closest to each cluster centroid, and each patch within a single feature is sampled from a unique case. Patches are 289 × 289 pixels obtained at 10X, with scale bar in lower right showing length of 100 micrometers.

**Table 1 Significance of adding machine-learned features to baseline clinicopathologic variables**

|  | Temporal validation | External validation 1a | External validation 1b |
|---|---|---|---|
| Null model (age, sex, tumor grade, T-category, lymphatic invasion, venous invasion) | Reference | Reference | Reference |
| *P*-value for adding 5 machine-learned features | <0.0001 | 0.00032[a] | <0.0001 |

*P*-values for a likelihood ratio test comparing a null model containing baseline clinicopathologic variables to an alternative model containing the baseline clinicopathologic variables and 5 machine-learned features.
[a]planned primary analysis.

learned features alone and was evaluated on the temporal validation set (outcome data was not available for the external validation sets). Patients were split into low and high risk groups within each stage based on the model's predicted probability of LNM (see Methods). In Kaplan-Meier analysis (Fig. 3), this model provided significant risk stratification within both stage II and stage III cases (*p* < 0.001, log-rank test). For stage II cases the 5-year DSS survival rate was 0.855 [95% 0.789, 0.901] for the low risk group vs. 0.664 [95% CI: 0.588, 0.736] for the high risk group. For Stage III cases, the 5-year DSS was 0.697 [95% CI: 0.588, 0.783] for the low risk group vs. 0.493 [95% CI: 0.430, 0.554] for the high risk group.

Finally, to provide initial insights into the histologic patterns captured by the machine-learned features, pathologists reviewed exemplar image patches sampled at random (see Methods) for each machine-learned feature (Fig. 2). The feature most strongly associated with LNM in univariable analysis (feature #1) consisted predominantly of adipose tissue and inflammatory cells with rare tumor cells. The features identified with significant ORs in multivariable analysis in external validation (#3 and #4), exhibited predominantly moderately differentiated tumor with occasional presence of inflammation and fibrotic stroma (#3; OR = 1.64 [95%CI: 1.18, 2.27] *p* = 0.003, external validation set 1a) and predominantly high grade adenocarcinoma with scant stroma (#4; OR = 0.62 [95%CI: 0.44, 0.88], *p* = 0.007; external validation set 1a) (ORs in Table 2). Brief summary descriptions for the five features based on review of the sampled patches are

shown in Supplementary Table S2, and 25 additional patches from external validation set 1a from each cluster are featured in Supplementary Fig. S2–S6.

**Discussion**

In this work, we identify machine-learned histopathologic features of primary resection specimens that predict the presence of LNM in CRC. Notably, these features provide independent signal relative to known clinicopathologic variables. While the boost in predictive performance achieved via the addition of ML features to our baseline model is modest, it does indicate that there is additional signal for LNM prediction when combined with what is currently known and used (e.g., T-stage, grade, lymphovascular invasion, venous invasion, etc), representing an opportunity for future research to further understand the features and biology associated with LNM. Without implying that this model is immediately applicable for clinical use, there are at least two important clinical decisions related to this type of prediction. First, for prognostic risk stratification to help identify high risk patients with Stage II cancer who may benefit from adjuvant chemotherapy[2,3]. Second, for risk assessment in endoscopic resection of apparent T1 cancer where there lymph node sampling is typically not performed but risk of metastasis is still 7–15%[9,10]. An approach that adds information to that of baseline features alone may be able to help inform decisions about the need for additional lymph node sampling or treatment escalation

**Table 2 Multivariable odds ratios for the machine-learned features**

| Covariate | Temporal validation | | External validation 1a | | External validation 1b | |
|---|---|---|---|---|---|---|
| | OR [95% CI] | *p* | OR [95% CI] | *p* | OR [95% CI] | *p* |
| **Age** | | | | | | |
| < 60 | 1.00 (reference) | n/a | 1.00 (reference) | n/a | 1.00 (reference) | n/a |
| 60–69 | 0.48 [0.29, 0.79] | 0.004 | 0.58 [0.35, 0.96] | 0.033 | 0.51 [0.35, 0.74] | < 0.001 |
| 70–79 | 0.51 [0.32, 0.83] | 0.007 | 0.39 [0.23, 0.64] | < 0.001 | 0.41 [0.27, 0.61] | < 0.001 |
| > 80 | 0.51 [0.31, 0.85] | 0.009 | 0.38 [0.21, 0.67] | < 0.001 | 0.35 [0.22, 0.57] | < 0.001 |
| **Sex** | | | | | | |
| Male | 1.00 (reference) | n/a | 1.00 (reference) | n/a | 1.00 (reference) | n/a |
| Female | 1.52 [1.09, 2.13] | 0.014 | 1.01 [0.70, 1.47] | 0.953 | 0.91 [0.68, 1.23] | 0.545 |
| **Tumor Grade** | | | | | | |
| G1 | 1.00 (reference) | 1.00 (reference) | 1.00 (reference) | n/a | 1.00 (reference) | n/a |
| G2 | 0.72 [0.45, 1.15] | 0.166 | 1.09 [0.75, 1.57] | 0.65 | 0.78 [0.58, 1.04] | 0.094 |
| G3 | 1.18 [0.70, 1.99] | 0.534 | 1.66 [0.81, 3.41] | 0.169 | 1.29 [0.68, 2.43] | 0.44 |
| **T-Category** | | | | | | |
| T2 | n/a | n/a | n/a | n/a | 1.00 (reference) | n/a |
| T3 | 1.00 (reference) | n/a | 1.00 (reference) | n/a | 1.56 [1.14, 2.13] | 0.005 |
| T4 | 1.24 [0.81, 1.88] | 0.319 | 1.04 [0.60, 1.81] | 0.884 | 1.87 [1.11, 3.16] | 0.018 |
| **Lymphatic Invasion** | | | | | | |
| L0 | 1.00 (reference) | 1.00 (reference) | 1.00 (reference) | n/a | 1.00 (reference) | n/a |
| L1+ | 1.76 [1.16, 2.68] | 0.008 | 5.48 [3.36, 8.96] | < 0.001 | 4.06 [2.76, 5.96] | < 0.001 |
| **Venous Invasion** | | | | | | |
| V0 | 1.00 (reference) | 1.00 (reference) | 1.00 (reference) | n/a | 1.00 (reference) | n/a |
| V1+ | 1.85 [1.12, 3.04] | 0.016 | 1.58 [0.83, 2.99] | 0.163 | 1.53 [0.88, 2.66] | 0.132 |
| **Machine-learned Features** | | | | | | |
| 1 | 1.88 [1.33, 2.68] | < 0.001 | 1.31 [0.47, 3.64] | 0.611 | 1.54 [0.67, 3.54] | 0.31 |
| 2 | 0.50 [0.20, 1.29] | 0.154 | 0.59 [0.32, 1.08] | 0.086 | 0.66 [0.42, 1.05] | 0.078 |
| 3 | 2.18 [1.48, 3.21] | < 0.001 | 1.64 [1.18, 2.27] | 0.003 | 1.45 [1.13, 1.86] | 0.003 |
| 4 | 0.80 [0.60, 1.07] | 0.13 | 0.62 [0.44, 0.88] | 0.007 | 0.63 [0.46, 0.85] | 0.003 |
| 5 | 0.87 [0.58, 1.32] | 0.516 | 1.04 [0.58, 1.86] | 0.907 | 0.77 [0.49, 1.21] | 0.264 |

Odds ratios (OR) from multivariable logistic regression model baseline clinicopathologic variables (age, sex, tumor grade, T-category, lymphatic invasion, venous invasion) and 5 machine-learned features.

**Table 3 AUROC for LNM prediction**

| Model | Temporal validation | External validation 1a | External validation 1b |
|---|---|---|---|
| Clinical | 0.667 [0.626, 0.708] | 0.716 [0.674, 0.762] | 0.719 [0.684, 0.752] |
| Clinical + ML | 0.715 [0.674, 0.753] | 0.740 [0.701, 0.780] | 0.738 [0.705, 0.770] |
| Delta | 0.048 [0.027, 0.069] | 0.024 [−0.001, 0.047] | 0.019 [0.000, 0.037] |

AUROCs for LNM predictions for logistic regressions with various feature sets. *Clinical* baseline clinicopathologic variables (age, sex, tumor grade, T-category, lymphatic invasion, venous invasion). *Clinical + ML* baseline clinicopathologic variables plus 5 machine-learned features. *Delta* the difference between Clinical + AI and Clinical. 95% confidence intervals computed via bootstrapping.

in some cases. While the lack of Stage I cases in our training data limits our ability to directly evaluate this potential in our study, it remains an intriguing area for future work.

In this work our goal was to maximize performance of an LNM prediction model when clinicopathologic and machine-learned features were combined. To this end, we decided a priori to include baseline clinicopathologic variables in our multivariate logistic regression model when selecting the machine-learned features. To evaluate the effectiveness of this approach, we also performed machine-learned feature selection without controlling for baseline clinicopathologic variables. Notably, in this negative control experiment, addition of machine-learned features selected in this manner did not increase performance over the clinicopathologic variables alone on the external validation sets (Supplementary Table S5). This supports the value of controlling for established clinicopathologic variables when attempting to both learn novel features and maximize overall predictive value of deep learning-based approaches.

Our work builds on and corroborates previous work investigating the ability of deep learning-based models to predict cancer metastasis from primary tumor tissue. Brinker et al. demonstrated that deep learning-extracted features could be used to predict sentinel LNM in melanoma, but also demonstrated that these features did not add value in combination with the baseline variables[17]. Kiehl et al. reported the performance of a deep learning-based model in prediction of LNM in colorectal cancer, and were able to show on their internal validation set that this model did improve predictive performance when added to baseline clinical features in a logistic regression model. However, they did not control for these baseline clinical features when training their deep learning model, and on their external validation set (TCGA) the deep learning model did not add value over the baseline features alone[18]. Similarly, Brockmoeller et al. demonstrated the development of a deep learning model for prediction of LNM in early colorectal cancer, and showed good performance that remained when combined with baseline models on their internal validation set, but they did not have an external dataset to assess generalizability[19]. To expand upon these prior efforts and address the issue of relearning established features, we control for known features during feature selection and

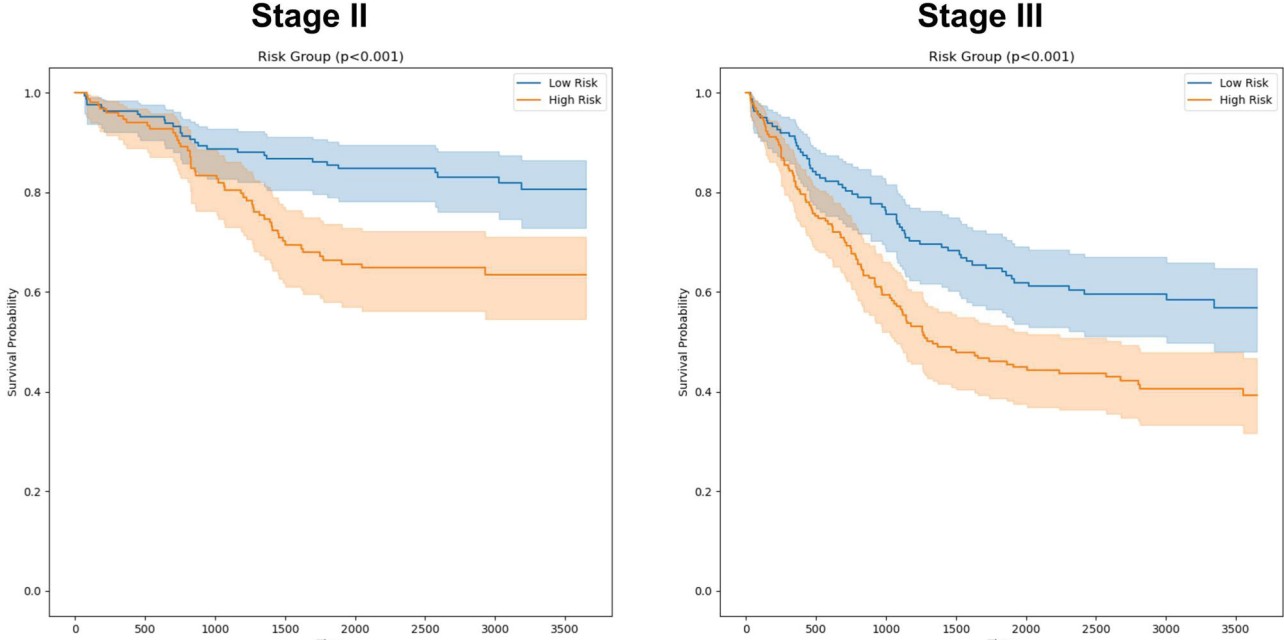

**Fig. 3 Kaplan-Meier Curves for LNM risk groups (temporal validation set).** Kaplan-Meier Curves for disease specific survival (DSS) amongst stage II and stage III cases, respectively. A logistic regression model using the five machine-learned features identified for LNM prediction was fit on the development cohort for predicting DSS. Risk groups were defined within each stage by binarizing using the median LNM model score for the most recent 5-years of the development cohort (2003 to 2007). The resulting regression models and risk group thresholds were then evaluated on the temporal validation set. The logistic regression model provided significant risk stratification within both node-negative and node-positive disease groups, suggesting the potential of such a model to aid in improving prognostication and therapeutic decision making.

demonstrate that this approach can generalize to an external dataset, and even to an external cohort comprising different characteristics than those in the training cohort (e.g., stage I/IV cases, T2 disease). Additionally we have demonstrated good performance while using a pretrained deep learning model to generate fixed embeddings (without requiring fine tuning on our specific task), substantially reducing the computational cost to develop such an algorithm[29,30].

The deep learning-extracted feature most strongly associated with LNM in our study (cluster #1) consisted of inflammatory cells and adipose tissue with occasional tumor cells. Interestingly, this finding is consistent with a recent finding from Brockmoeller et al of "inflamed adipose tissue" as a highly predictive feature for LNM presence[19]. Our study of T3 and T4 cases complements their study of T1 and T2 cases and further establishes inflamed fat as a feature that is not simply associated with depth of invasion. Additionally, this feature shares intriguing similarity with the tumor adipose feature (TAF) previously identified as predictive of disease-specific mortality in colorectal cancer[16], albeit with less predominant tumor cell involvement than TAF. While understanding the interaction of adipose tissue, inflammation, and tumor and potential biological underpinnings in these different contexts warrants further investigation, these findings provide compelling corroboration of one another as an important feature associated with morbidity in colorectal cancer. Additionally, machine-learned feature clusters as used in this study may comprise a diversity of individual morphological features, such that precise definitions of features that apply to all patches might not be possible. The potential to further define the most relevant morphological findings is an important future direction for this type of work. It is our hope that additional research into these emerging associations may yield insights into pathogenesis of LNM that are yet to be characterized and whose discovery may provide important avenues for improvement of clinical management.

One interesting, unanticipated challenge identified in this work was the confounding between scan date and LNM positivity (see Methods). For the development cohort, LNM negative cases were predominantly scanned before positive cases (because scanning proceeded by cancer stage). Early in model development we were achieving very high AUROC on the development set, and upon investigation discovered that the model had apparently learned an association between scan dates and lymph node status with cases scanned before a certain date receiving uniformly lower probability of metastasis. After investigating the scanner settings, image metadata, and viewing representative slides before and after this cutoff date, we were unable to identify a difference in these scans that might explain the observations, but clearly there was some confounding feature(s) learned by the model, a possibility shown previously[31]. The ability of neural networks to learn confounding associations has been well-documented, including computer vision applications in healthcare settings[32,33], and weakly-supervised settings such as the present work may be at increased susceptibility to this issue due to relatively low signal-to-noise ratio. Our findings underscore the importance of the data ingestion process as it relates to model development overall, highlighting that care should be taken to avoid the introduction of spurious associations.

Our study has several limitations. First, the data that was used for training and model development included only Stage II and III cases, which potentially limits the generalizability of our model outside of these stages. As such, our primary analysis focused on evaluation of the subset of stage II and III cases available within the external validation set. However, in secondary analysis we observed that the machine-learned features also provided significant predictive signal when including all stages. Additionally, given the confounding association between T-category and LNM amongst stage II and III cases (e.g., by definition, T1/T2 cases are never found in stage II), we excluded stage III T1 and T2 cases in

model development. As a result, our models may not capture the full association between T category and lymph node metastasis[18]. Additionally the lack of T1 cases and endoscopic resections in our dataset limited our ability to investigate model performance in the important use case detailed above of metastasis prediction when doing endoscopic resection for T1 cases. Second, we also note that Graph-Rise, the CNN architecture used for feature generation, while described publicly[25], is not available in off-the-shelf deep learning frameworks. Acknowledging this limitation, we repeated our approach and analyses using a commonly available model architecture, and found that good performance can be achieved using public models and datasets. Supplementary Fig. S7 shows top clusters from these models, and interestingly an adipose tissue predominant feature is one of two most predictive clusters in both cases, adding support to the above hypothesis that this has an important association with LNM in colorectal cancer. Third, our evaluation of this model as a risk stratification tool (in the DSS analysis) is limited as this is a retrospective study, and treatment pathways present an important confounding factor that is difficult to control for, including potential differences in neoadjuvant and adjuvant therapy. Though treatment guidelines within stage II and within stage III colorectal cancer cohorts are fairly uniform, at least some variability in treatment likely exists in the real world. Fourth, while our methodological approach does yield inherent interpretability to our predictive model and as mentioned we do add to a growing body of literature regarding the important of inflamed adipose tissue, an extensive interpretability analysis is beyond the scope of the present work which focuses on controlling for known variables when selecting deep learning embeddings to maximize model performance. Further exploration of the morphological features may be the subject of further work.

In summary, we developed and evaluated a method for generating machine-learned histomorphological features that provide novel signal for LNM in colorectal cancer. We showed that the machine-learned features produced by our method were significantly associated with LNM after controlling for known clinicopathologic variables in an external validation set. We also show that a model incorporating these features provides risk stratification for disease specific survival for patients with and without identified metastasis in a temporal validation set. These results support the potential value in further refinement and validation of our method for clinical risk stratification for colorectal cancer as well as exploring our proposed methodology in the context of other important prediction tasks in pathology and oncology.

## Data availability

This study utilized archived anonymized pathology slides, clinicopathologic variables, and outcomes from the Institute of Pathology and the Biobank at the Medical University of Graz and Stanford University. Interested researchers should contact K.Z. to inquire about access to Biobank Graz data and J.S. to inquire about access to Stanford University data; reasonable requests for research use will be considered and require ethics review prior to access. The source data used to generate Fig. 3 is available as Supplementary Data 2.

## Code availability

In this work we use pre-trained deep learning models from 3 different approaches (Graph-RISE, BiT, SimCLR) to produce embeddings, and show that machine-features derived from these embedding achieve similar performance in LNM prediction. The BiT model has been open sourced and is available on TFHub (https://tfhub.dev/google/bit/s-r50x1/1). Code for generating and evaluating the machine-learned features while controlling for baseline features is available on GitHub (https://github.com/Google-Health/google-health/tree/master/colorectal_lymph_node_metastasis_prediction)[34]. Code for pretraining a SimCLR model is available at (https://github.com/google-research/simclr).

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

## Acknowledgements

Google LLC and Verily Life Sciences funded this work. The authors would like to acknowledge the Google Health Pathology and labeling software infrastructure teams and in particular Yuannan Cai for software infrastructure support and data collection. We also appreciate the input of Boris Babenko and Dale Webster for their feedback on the manuscript. Last but not least, this work would not have been possible without the support of Biobank Graz, Dr. Christian Guelly, the efforts of the slide digitization teams at the Medical University Graz and Stanford University, and the participation of additional pathologists who reviewed the cases for quality control or to annotate tumor.

## Author contributions

J.D.K. performed the majority of the machine learning development and validation with input from D.F.S, E.W., P.-H.C.C., and Y.L. I.F.-A., T.B. M.P., R.R., H.M., K.Z., P.R., S.G., T.M., and J.S. collected and performed quality control for the data and reviewed pathology images. G.S.C., L.H.P., and C.H.M., obtained funding for data collection and analysis, supervised the study, and provided strategic guidance. J.K., D.F.S., and E.W. prepared the manuscript with input from all authors. D.F.S. and E.W. contributed equally.

## Competing interests

This study was funded by Google LLC. J.D.K., S.A., F.T., G.S.C., L.H.P., C.H.M., Y.L., P.-H.C.C., D.F.S., and E.W. are current or past employees of Google LLC and own Alphabet stock. I.F.-A. and T.B. are consultants of Google LLC. M.P., R.R., H.M., K.Z., P.R., S.G., T.M., and J.S. declare no competing interests.
