## [Peer Review File · Communications Medicine]

Reviewers' comments:

Reviewer #1 (Remarks to the Author):

The authors discuss a method to predict lymph node metastasis from primary tumor using deep learning. This is a non-trivial task, as essentially the imaging data which would be used for this is not available (and can be missing because one does not know what lymph nodes to sample, if any). The method is compared with standard clinicopathological features, and the authors show, that when the method is properly designed the deep learning imaging features add to the model performance.

The study itself is well-designed and sound, and the authors take care to provide an independent validation set. This is especially important since deep learning models are very sensitive to out-of-distribution samples. Nevertheless, the method itself is not very novel nor surprising. One primary point of concern is that both the clinicopathological model, as the model designed by the authors have a very low predictive value and the method seems to add little. For instance on the external validation set 1a the authors show an improvement of 0.024 in AUC and even less on 1b (Table 3).

While that by itself is an interesting result, it does not appear to be practice changing, while from a methodological perspective well-designed. One particular question one must be able to answer is how this would change treatment decisions. Are there cases which would not achieve different treatment when evaluated prospectively? This is especially important given the low baseline performance, and the low additivity of the features.

Reviewer #2 (Remarks to the Author):

In their manuscript „Predicting lymph node metastasis from primary tumor histology and clinicopathologic factors in colorectal cancer” Krogue et al. describe a how they use image features derived from histopathologic images together with clinicopathologic parameters to predict the presence of lymph node metastasis (LNM) in patients with colorectal cancer. While the idea is interesting, it is not novel and original and the authors themselves quote various publications with similar approaches. So for this manuscript to add additional value and be of interest to the readership of Communications Medicine, the results of their prediction would have to be excellent to outstanding. Which they are not. Specifically I have the following concerns:

1. Performance

While the authors demonstrate the possibility to predict the presence of LNM using features generated from different deep learning models (DLMs), the performance is only mediocre. In the abstract an AUC of only 0.638 for LNM detection is reported. Although the authors report a statistically significant increase of the AUC when clinicopathologic variables are combined with the machine learning features, AUC values still remain relatively low. In my opinion statistical significance here stems from a rather high case number making a very small effect statistically significant somehow. It should be noted, that the authors write AUC without giving any explanation of what this abbreviation stands for (area under the curve) and which AUC (Of the receiver operator characteristic? Of the precision recall curve? Of any other

curve?) is meant. If I assume, the area under the receiver operator characteristic (AUROC) is meant, this might not be the best metric after all (see for example Kleppe ESMO open 2022). The AUROC tends to overestimate a models performance and a number of other metrics (accuracy, sensitivity, specificity, precision, AUPRC, F1-Score, etc.) and visualizations (cross tables, ROC and PRC curves, etc.) are desperately needed.

2. Interpretation

I'm not really familiar with the concept of "temporal validation". As I understand it, these cases are taken from the same cohort as the training set, but it is arbitrarily split at a certain time point. I'm not sure, if this is enough to constitute a meaningful difference of these two data sets. This would explain why the biggest effect could be observed in the temporal validation set and external validation does not work equally well. It is somewhat surprising to me that in the temporal validation set gender has a similar effect on LNM as lymphatic invasion, which pathologists would suspect to be the strongest indicator of LNM. How do the authors explain this?

Furthermore, it would be really interesting to explore the morphological features systematically in much more detail. Unfortunately, this cannot be done from the few and small image patches (without scale bars) that have been provided. A much more detailed analysis and presentation would be warranted. For example, it would be interesting to see if lymphatic invasion(s) would have been caught on some patches and how predictive of LNM these patches would have been.

3. Miscellaneous

The authors should be commended for their open communication of the unexplainable "bug" which led to a very high performance on cases scanned prior to a certain date. However, it worries me that the underlying issue could not be identified. We had a similar problem, where an unscheduled scanner calibration would mess up our results. Why can the slides not simply be re-scanned? If the issues are resolved afterwards, this would make the results much more trustworthy. Unfortunately, I cannot check out the code as it has not been made available as of the time of this review.

Apart from not providing the computer code, the manuscript lacks a section on the "statistical methods". This should be standard to understand for example whether the correct tests have been used and the right assumptions have been made. In that regard, it is highly unusual to crop axes on Kaplan-Meier Curves as done in Figure 3. This visually amplifies the difference in the curves and therefore gives the unexperienced reader a false impression of the effect of the model. Also, Kaplan-Meier Curves from the external validation sets would be needed.

Furthermore, the manuscript lacks important detail on the hyperparameters of the deep learning models and the necessary soft- and hardware which was used to generate the results.

Dear Editors,

Thank you for your careful consideration of our manuscript. I appreciate the comments and would like to respond to some of the concerns expressed:

Reviewer 1 comments

The authors discuss a method to predict lymph node metastasis from primary tumor using deep learning. This is a non-trivial task, as essentially the imaging data which would be used for this is not available (and can be missing because one does not know what lymph nodes to sample, if any). The method is compared with standard clinicopathological features, and the authors show, that when the method is properly designed the deep learning imaging features add to the model performance.

The study itself is well-designed and sound, and the authors take care to provide an independent validation set. This is especially important since deep learning models are very sensitive to out-of-distribution samples. Nevertheless, the method itself is not very novel nor surprising. One primary point of concern is that both the clinicopathological model, as the model designed by the authors have a very low predictive value and the method seems to add little. For instance on the external validation set 1a the authors show an improvement of 0.024 in AUC and even less on 1b (Table 3).

While that by itself is an interesting result, it does not appear to be practice changing, while from a methodological perspective well-designed. One particular question one must be able to answer is how this would change treatment decisions. Are there cases which would not achieve different treatment when evaluated prospectively? This is especially important given the low baseline performance, and the low additivity of the features.

Our responses

R1.1 “The method itself is not very novel nor surprising”

- We would counter that our specific approach does include important novel elements. By treating clusters of deep learned patch embeddings as covariates with known variables, our approach controls for these known variables during feature selection. As such, it offers potential advantages over published deep learning approaches that may directly learn known risk-features or that deliberately use known features for the predictions. This also may be an important factor contributing to why previous work has not shown an improvement of the deep learning approach over known features in lymph node metastasis (LNM) prediction on external datasets. To our knowledge, this method of feature generation/selection has not been done before, and we demonstrate that this does indeed provide a performance boost over known baseline variables that generalizes to an external dataset.

R1.2 “One primary point of concern is that both the clinicopathological model, as the model designed by the authors have a very low predictive value and the method seems to add little. For instance on the external validation set 1a the authors show an improvement of 0.024 in AUC and even less on 1b (Table 3).”

- While we agree the marginal predictive value over baseline variables is modest, it does indicate that there is additional signal for LNM prediction outside of what is currently known and used (e.g., T-stage, grade, lymphovascular invasion, venous invasion, etc). We consider that to be quite surprising and an exciting opportunity for future clinical research. Such improvement over the baseline is perhaps not dissimilar from addition of individual risk factors in the development of the Framingham cardiovascular risk score, whereby addition of each individual factor may only increase performance by a small amount, but they are included in the model because the summative effect of all factors improves patient risk stratification. Additionally, on the methodological aspect, the use of visually similar embedding clusters in our approach enabled interpretability insights which support a growing and intriguing body of literature indicating that tumor adipose feature (TAF) may be an important independent predictor of tumor behavior and prognosis.

R1.3 “One particular question one must be able to answer is how this would change treatment decisions”

- We agree that our present work is not yet ready for clinical deployment, but rarely does a research advance have direct clinical implications without multiple validation studies. In this regard, there are at least two clinical scenarios for which this work has implications. First, the prognostic risk stratification for Stage II cases provided by the features identified in our approach suggests the potential for future validation and incorporation into adjuvant therapy decisions for Stage II patients - a challenging decision in some cases for which even modest improvements over known risk factors alone can be useful. Second, upon endoscopic resection of T1 cancer there is not an opportunity for lymph node sampling. A model with any performance improvement over baseline features may provide clinical utility by informing decisions about additional lymph node sampling or treatment escalation in some cases. While the lack of Stage I cases with clinical outcomes data limited our ability to directly evaluate this potential in our study, it remains an intriguing area for future work and represents one example of how this approach could influence treatment decisions given further validation. We can emphasize these points further in the discussion as useful.
- Beyond changing clinical treatment, we hope that this work may motivate and inform future discoveries as it indicates there remain important features regarding the pathogenesis of LNM that are yet to be characterized and whose discovery may provide important avenues for incorporation into clinical practice.

In their manuscript, "Predicting lymph node metastasis from primary tumor histology and clinicopathologic factors in colorectal cancer" Krogue et al. describe how they use image features derived from histopathologic images together with clinicopathologic parameters to predict the presence of lymph node metastasis (LNM) in patients with colorectal cancer. While the idea is interesting, it is not novel and original and the authors themselves quote various publications with similar approaches. So for this manuscript to add additional value and be of interest to the readership of Communications Medicine, the results of their prediction would have to be excellent to outstanding. Which they are not. Specifically I have the following concerns:

1. Performance

While the authors demonstrate the possibility to predict the presence of LNM using features generated from different deep learning models (DLMs), the performance is only mediocre. In the abstract an AUC of only 0.638 for LNM detection is reported. Although the authors report a statistically significant increase of the AUC when clinicopathologic variables are combined with the machine learning features, AUC values still remain relatively low. In my opinion statistical significance here stems from a rather high case number making a very small effect statistically significant somehow. It should be noted, that the authors write AUC without giving any explanation of what this abbreviation stands for (area under the curve) and which AUC (Of the receiver operator characteristic? Of the precision recall curve? Of any other curve?) is meant. If I assume, the area under the receiver operator characteristic (AUROC) is meant, this might not be the best metric after all (see for example Kleppe ESMO open 2022). The AUROC tends to overestimate a model's performance and a number of other metrics (accuracy, sensitivity, specificity, precision, AUPRC, F1-Score, etc.) and visualizations (cross tables, ROC and PRC curves, etc.) are desperately needed.

2. Interpretation

I'm not really familiar with the concept of "temporal validation". As I understand it, these cases are taken from the same cohort as the training set, but it is arbitrarily split at a certain time point. I'm not sure, if this is enough to constitute a meaningful difference of these two data sets. This would explain why the biggest effect could be observed in the temporal validation set and external validation does not work equally well. It is somewhat surprising to me that in the temporal validation set gender has a similar effect on LNM as lymphatic invasion, which pathologists would suspect to be the strongest indicator of LNM. How do the authors explain this?

Furthermore, it would be really interesting to explore the morphological features systematically in much more detail. Unfortunately, this cannot be done from the few and small image patches (without scale bars) that have been provided. A much more detailed analysis and presentation would be warranted. For example, it would be interesting to see if lymphatic invasion(s) would have been caught on some patches and how predictive of LNM these patches would have been.

3. Miscellaneous

The authors should be commended for their open communication of the unexplainable “bug” which led to a very high performance on cases scanned prior to a certain date. However, it worries me that the underlying issue could not be identified. We had a similar problem, where an unscheduled scanner calibration would mess up our results. Why can the slides not simply be re-scanned? If the issues are resolved afterwards, this would make the results much more trustworthy. Unfortunately, I cannot check out the code as it has not been made available as of the time of this review.

Apart from not providing the computer code, the manuscript lacks a section on the “statistical methods”. This should be standard to understand for example whether the correct tests have been used and the right assumptions have been made. In that regard, it is highly unusual to crop axes on Kaplan-Meier Curves as done in Figure 3. This visually amplifies the difference in the curves and therefore gives the unexperienced reader a false impression of the effect of the model. Also, Kaplan-Meier Curves from the external validation sets would be needed.

Furthermore, the manuscript lacks important detail on the hyperparameters of the deep learning models and the necessary soft- and hardware which was used to generate the results.

Our responses

R2.1 “While the idea is interesting, it is not novel and original and the authors themselves quote various publications with similar approaches. So for this manuscript to add additional value and be of interest to the readership of Communications Medicine, the results of their prediction would have to be excellent to outstanding. Which they are not”

- Please see our response to R1.1 and R1.2 above.

R2.2 “While the authors demonstrate the possibility to predict the presence of LNM using features generated from different deep learning models (DLMs), the performance is only mediocre. In the abstract an AUC of only 0.638 for LNM detection is reported. Although the authors report a statistically significant increase of the AUC when clinicopathologic variables are combined with the machine learning features, AUC values still remain relatively low. In my opinion statistical significance here stems from a rather high case number making a very small effect statistically significant somehow.”

- We agree that performance of the model remains modest, which is unsurprising as this is a very difficult task that human experts are unable to perform using histopathology alone. Additionally the multivariable odds ratio analysis indicates that the predictive value of the machine learned features is similar in magnitude to that of several established risk associated features such as “T” category and venous invasion. Our goal in this research is to develop a quantitative model that allows improved prediction over known clinicopathologic features alone, and to generate hypotheses regarding what may be important besides these known features (e.g., TAF). See also response to R1.2 above.

R2.3 “It should be noted, that the authors write AUC without giving any explanation of what this abbreviation stands for (area under the curve) and which AUC (Of the receiver operator characteristic? Of the precision recall curve? Of any other curve?) is meant. If I assume, the area under the receiver operator characteristic (AUROC) is meant, this might not be the best metric after all (see for example Kleppe ESMO open 2022). The AUROC tends to overestimate a models performance and a number of other metrics (accuracy, sensitivity, specificity, precision, AUPRC, F1-Score, etc.) and visualizations (cross tables, ROC and PRC curves, etc.) are desperately needed.”

- AUC in this work does indeed reflect area under the receiver operator characteristic (AUROC) and we have updated our manuscript to define this and use AUROC in place of AUC throughout. We can also add accuracy, sensitivity, and specificity at reasonable model setpoints.

R2.4 “I’m not really familiar with the concept of “temporal validation”. As I understand it, these cases are taken from the same cohort as the training set, but it is arbitrarily split at a certain time point. I’m not sure, if this is enough to constitute a meaningful difference of these two data sets.”

- Our motivation for use of temporal validation includes its reference and description in the TRIPOD checklist [1] and prior work which defines it as evaluating “the performance of a model on subsequent patients from the same centre(s)” [2] or using individuals “from the same institution in a different, usually later, time period” [3]. The TRIPOD publication further describes temporal validation as a “stronger design for evaluating model performance” than randomly splitting (see Figure 3 of ref [1]). As such, we feel the use of both temporal validation and an additional, external dataset represent a particular strength of this work in providing thoughtful and rigorous evaluation. We have also added citations regarding these references for temporal validation in the manuscript.
 - [1] Moons et al. Transparent Reporting of a multivariable prediction model for Individual Prognosis Or Diagnosis (TRIPOD): Explanation and Elaboration. *Annals of Internal Medicine* 2015
 - [2] Altman et al. Prognosis and prognostic research: validating a prognostic model. *BMJ* 2009
 - [3] Moons et al. Risk prediction models: II. External validation, model updating, and impact assessment. *Heart* 2012

R2.5 “It is somewhat surprising to me that in the temporal validation set gender has a similar effect on LNM as lymphatic invasion, which pathologists would suspect to be the strongest indicator of LNM. How do the authors explain this?”

- This is surprising to us as well, and as it is not replicated in the external datasets, we would suggest it is simply a peculiarity of that dataset.

R2.6 “Furthermore, it would be really interesting to explore the morphological features systematically in much more detail. Unfortunately, this cannot be done from the few and small image patches (without scale bars) that have been provided. A much more detailed analysis and presentation would be warranted. For example, it would be interesting to see if lymphatic invasion(s) would have been caught on some patches and how predictive of LNM these patches would have been.

- We are in agreement on this point. However, an extensive interpretability analysis is beyond the scope of the present work which focuses on controlling for known variables when selecting deep learning embeddings to maximize model performance. Further exploration of the morphological features may be the subject of further work.
- We will add patch size information to the figure legend for the relevant figures and could include additional patch examples for each cluster if useful.

R2.7 “The authors should be commended for their open communication of the unexplainable “bug” which led to a very high performance on cases scanned prior to a certain date. However, it worries me that the underlying issue could not be identified. We had a similar problem, where an unscheduled scanner calibration would mess up our results. Why can the slides not simply be re-scanned?”

- Thank you for your comment, and re-scanning is an interesting suggestion. Unfortunately, pulling, scanning, and re-archiving of these slides (over 25,000 individual slides which would require selective identification from the archive) given the nature of the multi-institutional partnership that enabled this work would be prohibitively time-consuming and expensive to coordinate.

R2.8 “Unfortunately, I cannot check out the code as it has not been made available as of the time of this review.”

- We are planning on open sourcing our code on Google Health’s github repository [link], which is written and waiting for approval. We expect this to be done before publication of our work.

R2.9 “the manuscript lacks a section on the “statistical methods”

- Thank you for this comment. We have updated our feature evaluation section within methods to specifically define the statistical methods section to address this concern.

R2.10 “it is highly unusual to crop axes on Kaplan-Meier Curves as done in Figure 3. This visually amplifies the difference in the curves and therefore gives the inexperienced reader a false impression of the effect of the model”

- We did not intend to exaggerate the model’s effect. The cropping was performed automatically by the method generating the plot, and we will update Figure 3 with the “uncropped” version.

R2.11 “Also, Kaplan-Meier Curves from the external validation sets would be needed”

- We would love to do this analysis, but unfortunately the external datasets do not contain the 5 year survival data that we would need for this, which is why we could only perform the analysis on the temporal validation set. We admit this as a limitation of our study, but as the KM curves are a secondary and not primary endpoint to our study, we consider this an acceptable limitation.

R2.12 “Furthermore, the manuscript lacks important detail on the hyperparameters of the deep learning models and the necessary soft- and hardware which was used to generate the results.”

- The training of our deep learning models is described in the supplementary methods, and the process by which embeddings from these models are clustered and then used in predictions is described in the Feature Generation section. This code will also be open sourced. We are happy to provide any other details that the reviewers feel are important.

We appreciate the thoughtful comments from reviewers and editors. Please find our point by point response below. We include the entirety of the reviewers' comments (numbering individual response points), and include our responses in bullets:

Reviewer #1:

R1.1

The authors discuss a method to predict lymph node metastasis from primary tumor using deep learning. This is a non-trivial task, as essentially the imaging data which would be used for this is not available (and can be missing because one does not know what lymph nodes to sample, if any). The method is compared with standard clinicopathological features, and the authors show, that when the method is properly designed the deep learning imaging features add to the model performance.

- Thank you for the positive summary comments

R1.2

The study itself is well-designed and sound, and the authors take care to provide an independent validation set. This is especially important since deep learning models are very sensitive to out-of-distribution samples. Nevertheless, the method itself is not very novel nor surprising.

- We would counter that our specific approach does include important novel elements that, to our knowledge, have not been previously described in the literature. By treating clusters of deep learned patch embeddings as covariates with known variables, our approach controls for these known variables during feature selection. As such, it is specifically designed to discover novel features, which is not the case for other published deep learning approaches that may directly learn known risk-features or that deliberately use known features for the predictions. This could also be an important factor contributing to why previous work has not shown an improvement of the deep learning approach over the use of known features in predicting lymph node metastasis (LNM) on external datasets. To our knowledge, this is a novel method of feature generation/selection that accounts for known risk factors, and we demonstrate that this does indeed provide a performance boost over the known factors alone in a manner that generalizes to an external dataset. We have edited the abstract to make our contribution clearer.

R1.3 One primary point of concern is that both the clinicopathological model, as the model designed by the authors have a very low predictive value and the method seems to add little. For instance on the external validation set 1a the authors show an improvement of 0.024 in AUC and even less on 1b (Table 3).

While that by itself is an interesting result, it does not appear to be practice changing, while from a methodological perspective well-designed. One particular question one must be able to answer is how this would change treatment decisions. Are there cases which would no achieve

different treatment when evaluated prospectively? This is especially important given the low baseline performance, and the low additivity of the features.

- While we agree the marginal predictive value over baseline variables is modest, it does indicate that there is additional signal for LNM prediction outside of what is currently known and used (e.g., T-stage, grade, lymphovascular invasion, venous invasion, etc). We consider that to be quite surprising and an exciting opportunity for future clinical research. Such improvement over the baseline is perhaps not dissimilar from addition of individual risk factors in the development of the Framingham cardiovascular risk score, whereby addition of each individual factor may only increase performance by a small amount, but they are included in the model because the summative effect of all factors improves patient risk stratification. Additionally, on the methodological aspect, the use of visually similar embedding clusters in our approach enabled interpretability insights which support a growing and intriguing body of literature indicating that tumor adipose feature (TAF) may be an important independent predictor of tumor behavior and prognosis^{2,3}.
- We agree that our present work is not yet ready for clinical deployment, but rarely does a research advance have direct clinical implications without multiple validation studies. In this regard, there are at least two clinical scenarios for which this work has implications. First, the prognostic risk stratification for Stage II cases provided by the features identified in our approach suggests the potential for future validation and incorporation into adjuvant therapy decisions for Stage II patients - a challenging decision in some cases for which even modest improvements over known risk factors alone can be useful. Second, upon endoscopic resection of T1 cancer there is not an opportunity for lymph node sampling. A model with any performance improvement over baseline features may provide clinical utility by informing decisions about additional lymph node sampling or treatment escalation in some cases. While the lack of Stage I cases with clinical outcomes data limited our ability to directly evaluate this potential in our study, it remains an intriguing area for future work and represents one example of how this approach could influence treatment decisions given further validation. We have emphasized these points in a new paragraph, which is now the third under the Discussion section
- Beyond changing clinical treatment, we hope that this work may motivate and inform future discoveries as it indicates there remain important features regarding the pathobiology of LNM that are yet to be characterized and whose discovery may provide important avenues for incorporation into clinical practice. We have added a sentence at the end of the fifth paragraph under the Discussion section to highlight this.

Reviewer #2:

R2.1

In their manuscript “Predicting lymph node metastasis from primary tumor histology and clinicopathologic factors in colorectal cancer” Krogue et al. describe a how they use image features derived from histopathologic images together with clinicopathologic parameters to predict the presence of lymph node metastasis (LNM) in patients with colorectal cancer. While the idea is interesting, it is not novel and original and the authors themselves quote various publications with similar approaches. So for this manuscript to add additional value and be of interest to the readership of Communications Medicine, the results of their prediction would have to be excellent to outstanding. Which they are not.

- Please see our responses to R1.2 and R1.3 above.

R2.2

Specifically I have the following concerns:

1. Performance

While the authors demonstrate the possibility to predict the presence of LNM using features generated from different deep learning models (DLMs), the performance is only mediocre. In the abstract an AUC of only 0.638 for LNM detection is reported. Although the authors report a statistically significant increase of the AUC when clinicopathologic variables are combined with the machine learning features, AUC values still remain relatively low. In my opinion statistical significance here stems from a rather high case number making a very small effect statistically significant somehow.

- We agree that performance of the model remains modest, which is unsurprising as this is a very difficult task that human experts are unable to perform using histopathology alone. Additionally the multivariable odds ratio analysis indicates that the predictive value of the machine learned features is similar in magnitude to that of several established risk associated features such as “T” category and venous invasion. Our goal in this research is to develop a quantitative model that allows improved prediction over known clinicopathologic features alone, and to generate hypotheses regarding what may be important besides these known features (e.g., TAF). See also response to R1.2 above.

R2.3

It should be noted, that the authors write AUC without giving any explanation of what this abbreviation stands for (area under the curve) and which AUC (Of the receiver operator characteristic? Of the precision recall curve? Of any other curve?) is meant. If I assume, the area under the receiver operator characteristic (AUROC) is meant, this might not be the best metric after all (see for example Kleppe ESMO open 2022). The AUROC tends to overestimate a models performance and a number of other metrics (accuracy, sensitivity, specificity, precision, AUPRC, F1-Score, etc.) and visualizations (cross tables, ROC and PRC curves, etc.) are desperately needed.

- AUC in this work does indeed reflect area under the receiver operator characteristic (AUROC) and we have updated our manuscript to define this and use AUROC in place

of AUC throughout. We have also added accuracy, sensitivity, specificity, positive predictive value, and negative predictive value of the baseline and baseline + AI models on each evaluation dataset using thresholds chosen to maximize harmonic mean of sensitivity and specificity (see the end of paragraph 3 under “Results” and Supplemental Table S2). To add additional clarity we have also added ROCs for the clinical vs clinical + ML models (Supplementary Figure S1 and also referenced in paragraph 3 under “Results”)

R2.4

2. Interpretation

I’m not really familiar with the concept of “temporal validation”. As I understand it, these cases are taken from the same cohort as the training set, but it is arbitrarily split at a certain time point. I’m not sure, if this is enough to constitute a meaningful difference of these two data sets. This would explain why the biggest effect could be observed in the temporal validation set and external validation does not work equally well.

- Our motivation for use of temporal validation includes its description in the TRIPOD checklist⁴, which is a standard reporting guideline listed in the EQUATOR network, and additional work which defines it as evaluating “the performance of a model on subsequent patients from the same centre(s)”⁵ or using individuals “from the same institution in a different, usually later, time period”⁶. The TRIPOD publication further describes temporal validation as a “stronger design for evaluating model performance” than randomly splitting (see Figure 3 of ref [4]). As such, the combined use of both temporal validation and an additional, external dataset represent a particular strength of this work in providing rigorous evaluation. We have also added citations regarding these references for temporal validation in the manuscript.

R2.5

It is somewhat surprising to me that in the temporal validation set gender has a similar effect on LNM as lymphatic invasion, which pathologists would suspect to be the strongest indicator of LNM. How do the authors explain this?

- This is surprising to us as well. It is possible that this is simply a peculiarity of that dataset as the finding is not replicated in our external datasets, but there is some precedent in the literature to suggest female sex is a risk factor of LNM⁷.

R2.6

Furthermore, it would be really interesting to explore the morphological features systematically in much more detail. Unfortunately, this cannot be done from the few and small image patches (without scale bars) that have been provided. A much more detailed analysis and presentation would be warranted. For example, it would be interesting to see if lymphatic invasion(s) would have been caught on some patches and how predictive of LNM these patches would have been.

- We are in agreement on this point, and have included an additional 25 patches from each machine learning feature in supplementary figures S2-S6. However, an extensive

interpretability analysis is beyond the scope of the present work which focuses on controlling for known variables when selecting deep learning embeddings to maximize model performance. Further exploration of the morphological features may be the subject of further work. We have emphasized these limitations in our limitations paragraph (paragraph 7 under Discussion).

- We have added patch size information to the figure legend for the relevant figures (Figure 2, Supplementary Figures S2-S7)

R2.7

3. Miscellaneous

The authors should be commended for their open communication of the unexplainable “bug” which led to a very high performance on cases scanned prior to a certain date. However, it worries me that the underlying issue could not be identified. We had a similar problem, where an unscheduled scanner calibration would mess up our results. Why can the slides not simply be re-scanned? If the issues are resolved afterwards, this would make the results much more trustworthy.

- Thank you for your comment. While re-scanning is an interesting suggestion, unfortunately, given the nature of the multi-institutional partnership that enabled this work, pulling, scanning, and re-archiving of these slides (over 25,000 individual slides which would require selective identification from the archive) would be prohibitively time-consuming.

R2.8

Unfortunately, I cannot check out the code as it has not been made available as of the time of this review.

- In this work we use pre-trained deep learning models from 3 different approaches (Graph-RISE, BiT, SimCLR) to produce embeddings, and show that machine-features derived from these embedding achieve similar performance in LNM prediction. The pre-trained BiT model has been open sourced and is available on TFHub (<https://tfhub.dev/google/bit/s-r50x1/1>). Code for pretraining a SimCLR model is available at (<https://github.com/google-research/simclr>). Our code for generating and evaluating the machine-learned features while controlling for baseline features will be made available on GitHub (<https://github.com/Google-Health/google-health>) at the time of publication. The reason for this is that there’s previously been instances of code leaking during review (ie, prior to publication). There was no practical way for the journal to intervene at that point, and this has increased the amount of scrutiny that code sharing like this receives.

R2.9

Apart from not providing the computer code, the manuscript lacks a section on the “statistical methods”. This should be standard to understand for example whether the correct tests have been used and the right assumptions have been made.

- Thank you for this comment. We have updated our feature evaluation section within methods to specifically describe the statistical methods and updating header to: “Statistical Evaluation of Features”.

R2.10

In that regard, it is highly unusual to crop axes on Kaplan-Meier Curves as done in Figure 3. This visually amplifies the difference in the curves and therefore gives the unexperienced reader a false impression of the effect of the model.

- We did not intend to exaggerate the model’s effect but appreciate the reviewer’s careful review and pointing this out. The cropping was performed automatically by the method generating the plot, and we have updated Figure 3 with the “uncropped” version.

R2.11

Also, Kaplan-Meier Curves from the external validation sets would be needed.

- We would love to do this analysis, but unfortunately the external datasets do not have paired survival data available, which is why we could only perform the analysis on the temporal validation set. We admit this as a limitation of our study, but as the KM curves are a secondary and not primary evaluation in this study, we consider this an acceptable limitation.

R2.12

Furthermore, the manuscript lacks important detail on the hyperparameters of the deep learning models and the necessary soft- and hardware which was used to generate the results.

- In this work we used Graph-RISE as a pretrained feature extractor and did not do any fine tuning. It’s training has been well-described in a previous work⁸. The BiT model used for additional evaluation was pretrained as well and it’s pretraining regimen is also well-described in previous work⁹. References to these works are included in the supplementary methods. The SimCLR model used for additional evaluation we did train ourselves, and we have detailed hyperparameter settings in the supplementary methods, including learning rate, batch size, patch sampling regimen, epochs, etc. We have added that it was trained on V2 TPU hardware to this section. The process by which embeddings from these models are clustered and then used in predictions is described in the Feature Generation section. This code will also be open sourced and additional details about software are provided in the code availability section. We are happy to provide any other details that the reviewers feel are important.

References

1. Kiehl, L., Kuntz, S., Höhn, J., Jutzi, T., Krieghoff-Henning, E., Kather, J. N., ... & Brinker, T. J. (2021). Deep learning can predict lymph node status directly from histology in colorectal cancer. *European Journal of Cancer*, 157, 464-473.
2. Wulczyn, E. et al. Interpretable survival prediction for colorectal cancer using deep learning. *NPJ Digit Med* 4, 71 (2021).
3. Brockmoeller, S., Echle, A., Laleh, N. G., Eiholm, S., Malmstrøm, M. L., Kuhlmann, T. P., ... & Kather, J. N. (2021). Deep Learning identifies inflamed fat as a risk factor for lymph node metastasis in early colorectal cancer. *The Journal of pathology*.
4. Moons et al. Transparent Reporting of a multivariable prediction model for Individual Prognosis Or Diagnosis (TRIPOD): Explanation and Elaboration. *Annals of Internal Medicine* 2015
5. Altman et al. Prognosis and prognostic research: validating a prognostic model. *BMJ* 2009
6. Moons et al. Risk prediction models: II. External validation, model updating, and impact assessment. *Heart* 2012
7. Ichimasa et al. Patient gender as a factor associated with lymph node metastasis in T1 colorectal cancer: A systematic review and meta-analysis. *Mol Clin Oncology* 2017
8. Juan, D.-C. et al. Graph-RISE: Graph-Regularized Image Semantic Embedding. *arXiv [cs.CV]* (2019).
9. Kolesnikov, A. et al. Big Transfer (BiT): General Visual Representation Learning. *Computer Vision – ECCV 2020* 491–507 (2020) doi:10.1007/978-3-030-58558-7_29.

Reviewers' comments:

Reviewer #1 (Remarks to the Author):

Thank you for your careful consideration of my and the other reviewers comments.

While I still have doubts about the impact given the minor improvement above baseline which requires the whole machinery of a ML pipeline versus a simple form, I believe my comments have been correctly addressed and this work is definitely worthy of publication.

Reviewer #2 (Remarks to the Author):

In their rebuttal, Krogue et al. address some of the issues raised in the initial revision of their manuscript. However, I do not really agree with their line of argumentation.

In R1.2 the authors state: "By treating clusters of deep learned patch embeddings as covariates with known variables, our approach controls for these known variables during feature selection.". How do the authors come to this conclusion? It would have been necessary a) to explain this in much more detail and b) prove this with experimental data. Both is still missing in my opinion. Even further, the results might even demonstrate the opposite: Feature 1 is described as "Adipose and inflammatory cells with occasional tumor cells". This is a feature which could indirectly correspond to the T stage as infiltration into the pericolonic adipose tissue is determining whether a tumor is pT2 or pT3. Features 2-4 describe different stages of tumor grading which we already know to be associated with more aggressive disease.

In R1.3 the arguments are rather subjective and with some I would respectfully disagree. In the stage II scenario for example, I would want a strong additional rationale to make any decision about further treatment, not a weak one. A good example of such parameters would be BRAF or MMR status. It is unclear if the slight performance improvement observed in the study would be able to reach the same level of additional information as these markers.

In R2.2 the authors state that prediction of lymph node metastasis "[...] is a very difficult task that human experts are unable to perform using histopathology alone.". Besides the fact that pathologists are rarely asked this question as lymph nodes are examined during routine pathology assessment anyways – the authors show no data and / or literature to back up this claim. In clinical reality, the detection of tumor infiltrates in lymphatic vessels for example is a strong indicator that lymph nodes might be positive as well.

In R2.6 Krogue et al. refer to additional example patches for each feature they have identified.

However, a substantial number of the patches shown does not even contain any tumorous tissue.

For example, in S2 you can find mostly fat and immune cells, but also hemorrhage (column 1 line 4), muscle tissue (column 5 line 5), and what seems to be healthy colonic crypts (column 3 line 4). In S3 you find muscle tissue (columns 2 and 3 line 4), mucus with vessels (column 5 lines 4 and 5) and ink for highlighting the resection margin (column 1 line 4). What is the authors explanation that a) there are no tumor cells on these patches and b) they do not correspond to the features as described in the paper?

In summary, my initial assessment has not substantially changed, and my concerns have not really been addressed. As I understand it, this is an appeal of a rejection. So, the arguments would have to hold up to an even higher level of scrutiny.

We appreciate the thoughtful comments from reviewers and editors. We include the entirety of the reviewers' comments (numbering individual response points), and include our responses in blue.

Reviewer #1:

R1.1

Thank you for your careful consideration of my and the other reviewers' comments.

While I still have doubts about the impact given the minor improvement above baseline which requires the whole machinery of a ML pipeline versus a simple form, I believe my comments have been correctly addressed and this work is definitely worthy of publication.

We very much appreciate your careful review and positive comments.

Reviewer #2:

R2.1

In their rebuttal, Krogue et al. address some of the issues raised in the initial revision of their manuscript. However, I do not really agree with their line of argumentation.

In R1.2 the authors state: "By treating clusters of deep learned patch embeddings as covariates with known variables, our approach controls for these known variables during feature selection.". How do the authors come to this conclusion? It would have been necessary a) to explain this in much more detail and b) prove this with experimental data. Both is still missing in my opinion. Even further, the results might even demonstrate the opposite: Feature 1 is described as "Adipose and inflammatory cells with occasional tumor cells". This is a feature which could indirectly correspond to the T stage as infiltration into the pericolonic adipose tissue is determining whether a tumor is pT2 or pT3. Features 2-4 describe different stages of tumor grading which we already know to be associated with more aggressive disease.

We apologize for the lack of clarity in the statement, "By treating clusters of deep learned patch embeddings as covariates with known variables, our approach controls for these known variables during feature selection." To clarify, the feature selection approach attempts to discover features that are associated with LNM after controlling for known variables through use of multivariate logistic regression. We have updated the feature selection section to provide additional clarity:

"Given a set of K candidate machine-learned features, a subset of machine-learned features was selected for inclusion in an LNM prediction model that combines both clinicopathologic variables and machine-learned features. Forward stepwise selection was employed for machine-learned feature selection with the baseline clinicopathologic variables included in the model throughout the process. In other words, we started with a multivariable logistic regression model for LNM that included only the set of known baseline variables. Candidate machine-learned features were then iteratively selected for inclusion in the multivariable logistic regression model. In each iteration, the candidate machine-learned feature that gave the largest increase in performance

(AUROC) when added to the model was selected. We measured performance on the development set over different values of K clusters (10, 25, 50, 100, 200), and for an increasing number of selected machine-learned features (1-10). The optimal configuration of K=200 and 5 selected machine-learned features was chosen based on development set performance and the observation of diminishing returns after selecting more than 5 machine-learned features.

The goal of this selection process was to identify a subset of machine-learned features that are associated with LNM after controlling for known clinicopathologic variables by including them in the multivariable model. The statistical evaluations used to evaluate this approach are described next.”

The reviewer also makes a good point about what happens in a negative control experiment without using our approach. We have reported results for such a control experiment in (Supplementary Table S4), where the results indicate that features selected *without* controlling for the baseline variables did not provide a performance improvement over the baseline variables alone in the external validation sets.

We appreciate the Reviewer's observation that Feature 1 “could indirectly correspond to the T-Stage”. While there is an association between Feature 1 and T-Stage ($p=0.032$), Feature 1 captures information relevant for LNM that is not captured by T-Stage alone: the AUC for T-Stage alone is 0.571 [0.548, 0.599], while the AUC for a logistic regression model containing both T-Stage and Feature 1 is 0.656 [0.622, 0.691].

Similarly, while Features 2-4 are associated with Grade visually, they capture information relevant for LNM that is not captured by Grade alone. The AUC for Grade alone is 0.568 [0.530, 0.605] while the AUC for a logistic regression model containing both Grade and Features 2-4 is 0.649 [0.612, 0.691].

R2.2

In R1.3 the arguments are rather subjective and with some I would respectfully disagree. In the stage II scenario for example, I would want a strong additional rationale to make any decision about further treatment, not a weak one. A good example of such parameters would be BRAF or MMR status. It is unclear if the slight performance improvement observed in the study would be able to reach the same level of additional information as these markers.

We appreciate the thoughtful comments on this topic and have updated the language in the manuscript to moderate any suggestion of immediate clinical impact which is not our intention. The discussion in the original R1.3 response was intended to communicate that decisions about adjuvant therapy remain challenging in many cases despite many known risk factors that influence the decision (PMID: 28399381) - and that as such, that having additional independent information provided could be useful. Our intention was not to suggest in any way that the machine learned features are necessarily as or more informative in isolation than any established risk factors.

R2.3

In R2.2 the authors state that prediction of lymph node metastasis “[...] is a very difficult task that human experts are unable to perform using histopathology alone.”. Besides the fact that pathologists are rarely asked this question as lymph nodes are examined during routine pathology assessment anyways – the authors show no data and / or literature to back up this claim. In clinical reality, the detection of tumor infiltrates in lymphatic vessels for example is a strong indicator that lymph nodes might be positive as well.

Thank you for this comment. We agree that it is difficult to specify just how accurately pathologists themselves can predict LNM directly from tumor tissue as it’s something not commonly asked, and we do recognize there are known risk factors for LNM in primary tissue (such as lymphatic invasion as you stated, which in our work we in fact demonstrate to have a clinically and statistically significant association with LNM [see Table 4])[1,2,3]. Our motivation in stating this in our response was to contextualize our work and results: i.e. this is a challenging task which experts do not routinely perform in clinical practice as opposed to a well-defined diagnosis that is routinely made by human experts. We are very careful in our paper to avoid making any claim regarding human performance in this task as we agree it is not well-defined.

R2.4

In R2.6 Krogue et al. refer to additional example patches for each feature they have identified. However, a substantial number of the patches shown does not even contain any tumorous tissue. For example, in S2 you can find mostly fat and immune cells, but also hemorrhage (column 1 line 4), muscle tissue (column 5 line 5), and what seems to be healthy colonic crypts (column 3 line 4). In S3 you find muscle tissue (columns 2 and 3 line 4), mucus with vessels (column 5 lines 4 and 5) and ink for highlighting the resection margin (column 1 line 4). What is the authors explanation that a) there are no tumor cells on these patches and b) they do not correspond to the features as described in the paper?

Thank you for your detailed review of the additional sample patches for the machine-learned features that were added to the supplement (S2-S6).

Regarding point a), the reason why there may not be tumor cells in many of the patches is that the region of interest (ROI) masks that were used to select patches for clustering were designed to include both tumor regions and small areas of non-tumor regions immediately adjacent to tumor. This was achieved by running the tumor detection model to create a tumor mask and then dilating the tumor mask to include adjacent non-tumor regions (~0.5mm or 2 patch widths). We have updated the “Region of Interest Selection” section in Methods to clarify and emphasize this point and to provide performance metrics for the tumor detection model itself.

Regarding point b), the some patches within a “feature cluster” may exhibit morphological findings that were not included in the summary descriptions because the notion of a machine-learned feature as a “cluster” in embedding space may capture a diversity of traditional morphological features (and this in turn may further vary with the number of clusters and the distance from cluster centroids for any given patch). Thus, the descriptions are intended to give a pathologist-provided summarization of the machine learned features rather than a precise and complete histopathological definition. We have added to the fourth paragraph of the Discussion section to raise the point about machine learned features not necessarily corresponding

perfectly to a limited set of morphological features and to note that the potential to further define the relevant aspects of a given feature is an interesting direction for future work.

R2.5

In summary, my initial assessment has not substantially changed, and my concerns have not really been addressed. As I understand it, this is an appeal of a rejection. So, the arguments would have to hold up to an even higher level of scrutiny.

We appreciate your careful review and hope that the additional information we have provided addresses your concerns.

References

1. Suh, J. H. *et al.* Predictors for lymph node metastasis in T1 colorectal cancer. *Endoscopy* **44**, 590–595 (2012).
2. Akishima-Fukasawa, Y. *et al.* Histopathological predictors of regional lymph node metastasis at the invasive front in early colorectal cancer. *Histopathology* **59**, 470–481 (2011).
3. Yamauchi, H. *et al.* Pathological predictors for lymph node metastasis in T1 colorectal cancer. *Surg. Today* **38**, 905–910 (2008).

REVIEWERS' COMMENTS:

Reviewer #2 (Remarks to the Author):

All my comments have been addressed. Thank you.